# Reduced Washing Cycle for Sustainable Mackerel (*Rastrelliger kanagurta*) Surimi Production: Evaluation of Bio-Physico-Chemical, Rheological, and Gel-Forming Properties

**DOI:** 10.3390/foods10112717

**Published:** 2021-11-06

**Authors:** Panumas Somjid, Worawan Panpipat, Ling-Zhi Cheong, Manat Chaijan

**Affiliations:** 1Food Technology and Innovation Research Center of Excellence, School of Agricultural Technology and Food Industry, Walailak University, Nakhon Si Thammarat 80160, Thailand; panumas.s14@gmail.com (P.S.); pworawan@wu.ac.th (W.P.); 2Zhejiang-Malaysia Joint Research Laboratory for Agricultural Product Processing and Nutrition, College of Food and Pharmaceutical Science, Ningbo University, Ningbo 315211, China; cheonglingzhi@nbu.edu.cn

**Keywords:** mackerel, surimi, washing, gel, quality, sustainability

## Abstract

Although dark muscle is currently the most important obstacle in marketing high-quality Indian mackerel (*Rastrelliger kanagurta*) surimi, reducing washing remains a challenge for long-term surimi production from this species. Herein, the impact of washing cycles (one (W1), two (W2), and three (W3) cycles) with a 1:3 mince to water ratio on the bio-physico-chemical properties, rheology, and gelling ability of mackerel surimi was evaluated. The yield, Ca^2+^-ATPase activity, TCA-soluble peptide, and myoglobin contents of surimi decreased as the number of washing cycles increased, while lipid removal, reactive SH content, and surface hydrophobicity of surimi increased. Surimi generated by W2 and W3 provided the same rheological patterns and Fourier-transform infrared spectroscopy (FTIR) spectra as unwashed mince, with the highest gel strength and whiteness, as well as the lowest expressible drip, thiobarbituric acid reactive substances (TBARS), and fishy odor. Sodium dodecyl sulfate-polyacrylamide gel electrophoresis (SDS-PAGE) demonstrated the presence of polymerized proteins stabilized by disulfide and other interactions. Using a scanning electron microscope, several concentrated dense areas and distributed pores generated by myofibrillar proteins gel networks were found. Surimi from W2 and W3 appeared to be of similar overall quality, however W2 had a larger yield. As a result of the evaluation of bio-physico-chemical, rheological, and gel-forming capabilities, as well as product yield, W2 may be the best option for producing high-quality surimi from Indian mackerel in a sustainable manner.

## 1. Introduction

Surimi demand in the global market has been driven by the increased desire for seafood analogs in recent decades, owing to various textural features, nutritional benefits, and easier consumable convenience [1]. In Thailand, the main white-fleshed fish used to manufacture surimi include bigeye snapper (*Priacanthus* spp.), threadfin bream (*Nemipterus* spp.), lizardfish (*Saurida* spp.), and goatfish (*Upeneus* spp.) [2]. Due to high demand, these fish species have been overfished, resulting in a downward trend in their populations over the last ten years (2010–2019) [3]. Pelagic dark-fleshed fish species have, on the other hand, remained abundant and stable in southern Thailand [4]. Consequently, dark-fleshed fish have received a lot of attention as prospective production alternatives. Indian mackerel (*Rastrelliger kanagurta*) is one of the most important pelagic fish in the Gulf of Thailand [3]. Unfortunately, this fish has been reported to have a high dark muscle content and a high level of unwanted elements (e.g., sarcoplasmic proteins, myoglobin, and lipids) [4]. As a result, producing high-quality surimi is still challenging.

The volume of water and the number of washing cycles required to produce high-quality surimi are influenced by the species of fish, the condition of the fish, and the desired product quality [5]. When impurities are eliminated, the color, texture, and odor of the finished product can be greatly enhanced. More mince to water ratio is commonly stated to offer a good result for gel properties, as it not only increases the cost of production, but also increases the amount of waste water and the loss of some myofibrillar protein [5]. It has been reported that 1 kg of surimi requires almost 15 vol. of water to produce. Surimi production’s extensive use of freshwater has had a severe influence on the environment, due to the discharge of untreated processing water [5]. As a result, there is a need for environmentally friendly surimi production with fewer washing cycles, not only to avoid pollution, but also to increase yield and minimize wash water content. Several studies have reported the use of washing solutions such as tap water [6], sodium chloride [7], sodium bicarbonate [8], carbonated water [9], and ozonized water [10] for mackerel surimi production. All of the above-mentioned washing media have been shown to improve quality parameters, but the results differ depending on the fish species. Still, after a safety concern, employing cold water as a chemical-free technique of washing minced fish for surimi manufacture should be a key element in the process. However, information on the effects of the number of washing cycles with water on bio-physico-chemical, rheological, and gelling properties of Indian mackerel surimi is scarce. With this in view, the goal of this research was to explore how a reduced number of washing cycles affected the quality of Indian mackerel surimi. This concept can serve as a basis for surimi industry sustainability.

## 2. Materials and Methods

### 2.1. Chemicals

All chemicals, such as sodium dodecyl sulfate (SDS), β-mercaptoethanol (βME), adenosine triphosphate (ATP), trichloroacetic acid (TCA), 5,5-dithio-bis (2-nitrobenzoic acid) (DTNB), and thiobarbituric acid (TBA), were purchased from Sigma Aldrich (St. Louis, MO, USA). All chemicals were of analytical grade.

### 2.2. Fish Sample

Indian mackerel (*R. kanagurta*), with an average weight of 60–70 g, were purchased from the market in Thasala, Nakhon Si Thammarat, southern Thailand. The fish were offloaded around 12 h after capture from the Khanom-Nakhon Si Thammarat Coast, packed in ice with a fish per ice ratio of 1:2 (*w*/*w*), and sent to the laboratory within 30 min. The fish was washed, gutted, filleted, and skinned immediately. The whole muscles were minced uniformly in a Panasonic MK-G20NR-W meat grinder (Osaka, Japan).

### 2.3. Surimi and Surimi Gel Preparation

To produce surimi with different washing cycles, minced fish was separated into three groups for washing: one-washing cycle (W1), two-washing cycles (W2), and three-washing cycles (W3). A mince to cold water (4 °C) ratio of 1:3 (*w*/*v*) was employed for repeatedly washing by gently swirling for 10 min/cycle in a cold environment (4 °C). Then, the washed mince was filtered through a nylon screen layer. After that, the mince was dewatered with a hydraulic press, and the moisture content of the unwashed mince and surimi was assessed using the AOAC method [11]. Unwashed mince had an initial moisture content of approximately 75%, while surimi had a moisture content of 78–80%. Using chilled water, the final moisture content of the unwashed mince and surimi was adjusted to an average of 80%, and the moisture level was determined again to ensure an 80% final moisture content. The difference between the weight of the original fish mince and the end mass of surimi was used to calculate the yield. Finally, sucrose and sorbitol, each at 4% *w*/*w*, were added into washed and unwashed mince as cryoprotectants, and frozen in a blast freezer. The samples were kept at −18 °C until they were needed (<a month). To produce the gels, frozen surimi and unwashed mince were brought to a temperature of 0–4 °C, where they could easily be cut into pieces. The surimi was then combined with 2.5% (*w*/*w*) salt using an HR2118/01 Philips blender (Philips Electronics (Thailand) Co., Ltd., Samut Prakan, Thailand) at low speed for 5 min to form the paste. Surimi paste was placed into a 2.5 cm diameter polyvinylidene chloride casing and firmly sealed. A two-step heating procedure was used for gelation, with temperatures of 40 °C for 30 min, and 90 °C for 20 min. Before analysis, all samples were chilled in ice water for 60 min and kept at 4 °C overnight [7].

### 2.4. Determination of Bio-Physico-Chemical Properties of Unwashed Mince and Surimi

The pH, Ca^2+^-ATPase activity, SH content, protein surface hydrophobicity, and TCA-soluble peptides were determined using the standard methods as described by Somjid et al. [9].

The pH was obtained using a pH meter (Cyberscan 500, Singapore) after homogenizing the sample with 10 vol of deionized water (*w*/*v*), using an IKA Labortechnik homogenizer (Selangor, Malaysia).

Ca^2+^-ATPase activity, SH content, and protein surface hydrophobicity were measured employing natural actomyosin (NAM) from unwashed mince and surimi. For the Ca^2+^-ATPase activity, NAM was diluted to 2.5–8 mg/mL with 0.6 M KCl (pH 7.0). A measure of 1 mL NAM solution was combined with 0.6 mL 0.5 M Tris-maleate (pH 7.0) and 1 mL 0.1 M CaCl_2_. To make up a total volume of 9.5 mL, deionized water was added. Then, a 0.5 mL solution of 20 mM ATP was added. After an 8-min incubation at 25 °C, the reaction was stopped by adding 5 mL cold 15% (*w*/*v*) TCA. After that, centrifugation (3500× *g*/5 min/25 °C) was performed using a RC-5B plus centrifuge (Sorvall, Norwalk, CT, USA), and the inorganic phosphate liberated in the supernatant was quantified. The activity of the Ca^2+^-ATPase was measured in moles of inorganic phosphate released/mg protein/min. A blank solution was prepared by adding chilled TCA prior to addition of ATP. For the SH content, the NAM sample (0.5 mL, 4 mg/mL) was combined with 4.5 mL of 0.2 M Tris-HCl buffer (pH 6.8) and 0.5 mL of 0.1% DTNB solution in a total volume of 7.5 mL. The absorbance was measured at 412 nm using a Shimadzu UV-2100 spectrophotometer (Shimadzu Scientific Instruments Inc., Columbia, MD, USA) after a 25-min incubation at 40 °C. By substituting 0.6 M KCl (pH 7.0) for the sample, a blank was created. The absorbance was used to calculate the SH content, which was reported as mol/10^8^ g protein, using the molar extinction of 13,600 M^−1^cm^−1^. The hydrophobicity of NAM dispersed in 20 mM phosphate buffer (pH 6) was measured by bromophenol blue sodium salt (BPB) for electrophoresis. To 1 mL of NAM suspension, 200 µL of 1 mg/mL BPB (in distilled water) was added, and mixed well. A control group was created without NAM. For 10 min, samples and controls were agitated at room temperature (26–28 °C). The absorbance of the supernatant was measured at 595 nm (A) using a Shimadzu UV-2100 spectrophotometer against a blank of phosphate buffer after centrifugation (2000× *g*/15 min/room temperature). The following formula was used to calculate the amount of BPB bound:(1)$$\mathrm{BPB}\;\mathrm{bound}\;\left(\mathrm{µg}\right)=200\;\mathrm{µg}\;\times \left[\frac{A_{\mathrm{control}}-{\;A}_{\mathrm{sample}}}{A_{\mathrm{control}}}\right]$$

For the TCA-soluble peptide, the sample was homogenized with 3 vol of 5% (*w*/*v*) TCA before being placed on ice for 1 h. The soluble peptides in the supernatant were measured and recorded as μmole tyrosine/g sample after centrifugation at 5000× *g* for 5 min.

Myoglobin was extracted using a cold 40 mM phosphate buffer, pH 6.8, according to Chaijan et al. [6]. The spectra in the visible and Soret regions were recorded using a Shimadzu UV-2100 spectrophotometer (Columbia, MD, USA). The following equations were used to calculate the proportions of the three myoglobin variants [12].
[Deoxymyoglobin] = −0.543P1 + 1.594P2 + 0.552P3 − 1.329(2)
[Oxymyoglobin] = 0.722P1 − 1.432P2 − 1.659P3 + 2.599(3)
[Metmyoglobin] = −0.159P1 − 0.085P2 + 1.262P3 − 0.520(4)
where P1 = A582/A525, P2 = A557/A525, and P3 = A503/A525.

Bligh and Dyer’s method [13] was used to extract lipids. The percentage of lipid reduction was calculated in comparison to the unwashed mince’s initial lipid content.

### 2.5. Determination of Rheological Properties

Following the method of Fukushima et al. [14], the rheological properties of unwashed mince and surimi pastes were measured using a HAAKE MARS 60 Rheometer (Thermo Fisher Scientific Inc, Yokohama, Japan). Approximately 0.5 g of pastes were spread on the sample holder under a 35 mm parallel plate geometry. The gap between the plate and the sample holder was fixed at 0.5 mm, and a thin layer of oil was added to prevent dehydration. Heating was performed at a constant frequency of 1 Hz and an amplitude strain of 2%, which was tithing the linear viscoelastic region. The temperature was swept from 10 to 90 °C at a rate of 2 °C per min, and the variations in rheological parameters, such as elastic modulus (G’), viscous modulus (G”), and tan δ, were recorded.

### 2.6. Determination of Gelling Properties

The standard methods reported by Chaijan et al. [15] were used to evaluate the texture (breaking force, deformation, and gel strength), expressible drip, whiteness, and thiobarbituric acid reactive substances (TBARS) of gels.

The breaking force (g) and deformation (mm) were determined with a spherical head plunger (dia 5 mm) pressed into the specimen (5-cm high/6 cm per min/60% compression) using a texture analyzer (Stable Micro Systems, Godalming, Surrey, UK). The gel strength (g·cm) was obtained by multiplying the breaking force with the deformation. For the expressible drip, a gel sample (0.5 cm thick) was weighed and placed between layers of Whatman filter paper No. 1 (two pieces at the top, and three pieces at the bottom). After compressing with the standard weight (5 kg) for 2 min, the sample was reweighed. Expressible drip was reported as percentage of sample weight. The gel color (*L**, a*, and *b**) was determined using a Hunterlab Miniscan/EX instrument, and the whiteness was calculated as follows:(5)$$\mathrm{Whiteness}=100-\sqrt{\left[{\left(100-L^{*}\right)}^{2}+a^{*2}+b^{*2}\right]}$$

For the TBARS assay, sample (0.5 g) was homogenized with 2.5 mL of a TBARS solution (0.375% TBA, 15% TCA, and 0.25 N HCl). The homogenate was heated in a boiling water bath (95–100 °C) for 10 min to develop a pink color, cooled with running tap water, and centrifuged at 3600× *g* at 25 °C for 20 min. The absorbance of the supernatant was measured at 532 nm. A standard curve was prepared using 1,1,3,3-tetramethoxypropane at concentrations ranging from 0 to 10 ppm. TBARS was calculated and reported as mg malondialdehyde equivalent/kg sample.

The fishy odor analysis of unwashed mince and surimi gels was performed following the experimental protocol approved by the Human Research Ethics Committee of Walailak University (WUEC-21-125-01). Gel samples (3-cm diameter and 1-cm thickness) were conditioned for 30 min at room temperature. The gel samples were assessed by 10 trained panelists by sniffing. All of them are graduate students who were trained to have rich professional knowledge on fishy odor. A line scale was used to rate the strength of the fishy odor (0–10, representing none-strong).

Protein patterns of unwashed mince, surimi, and their gels were studied using sodium dodecyl sulfate-polyacrylamide gel electrophoresis (SDS-PAGE) under non-reducing (without βME) and reducing (with βME) conditions using, the method of Laemmli [16]. Fourier transform infrared (FTIR) spectra of gels in the region of 400–4000 cm^−1^ were obtained using a Bruker INVENIO-S FTIR spectrometer (Bruker Co., Ettlingen, Germany), equipped with an attenuated total reflection (ATR) diamond crystal cell. A scanning electron microscopy (SEM) (GeminiSEM, Carl Ziess Microscopy, Oberkochen, Germany) was used to visualize the microstructure of gels [9].

### 2.7. Statistical Analysis

An analysis of variance was performed on the data. Duncan’s multiple-range test was used to compare the means. SPSS 8.0 for Windows (SPSS Inc., Chicago, IL, USA) was used to conduct the statistical analysis.

## 3. Results and Discussion

### 3.1. Yield

Surimi yields were in the 80–90% range across all washing processes (Table 1), and yields declined as the number of washing cycles increased (*p* < 0.05). Traditionally, cold water was used to eliminate contaminants from mince, improving its quality. Filleting, deboning, mincing, and washing reduce the percentage of raw material yield to surimi, which has an effect on surimi productivity [17]. The yield in surimi was reduced by eliminating sarcoplasmic protein, blood, pigment, and fat with increasing washing cycles in the current study, but the yield was still >80%. With increasing water-washing cycle up to 4-cycle, the yield gained from *Pangasius hypophthalmus* mince to surimi decreased dramatically [17].

### 3.2. Bio-Physico-Chemical Properties

#### 3.2.1. pH

The pH of unwashed mince was 6.31 at the beginning, and the pH of surimi increased significantly as the number of washing cycles increased (*p* < 0.05; Table 1). Surimi has a higher pH than unwashed mince, which could be attributed to the removal of water-soluble acidic compounds (e.g., free acidic amino acids and other endogenous acids) during the washing process. It has been reported that organic compounds, including acidic substances, can be removed from fish mince during the washing process [6,7,8]. According to Foegeding et al. [18], the ultimate pH is generally between 6.2 and 6.6. The pH of the surimi may be modified by the leaching efficiency, and related to the pH of the medium utilized [9]. However, the same medium was employed in this investigation, which was cold water. The only difference between the treatments was the washing cycle. As a result, the washing cycle can influence the pH of the surimi. During the washing process, both alkaline and acidic substances can be washed out of the minced fish, resulting in a new pH equilibrium in the washed mince [7,9,15].

#### 3.2.2. Lipid Reduction

For improved surimi quality, the lipid content must be reduced. Lipid can hinder gel formation, and lead to gel rancidity [7,19]. With increasing the washing cycle with water, lipid was observed to be reduced in sardine, mackerel, pink perch, and croaker surimi [20]. A lipid-stabilizing procedure was established to prevent the oxidation of residual lipid in mackerel surimi, which comprised the early addition of both lipid- and water-soluble antioxidants, the exclusion of extra sodium chloride, and the avoidance of oxygen [19]. Table 1 shows that, during washing, more than half of the lipid can be eliminated. More lipid was removed as the number of washing cycles increased. However, W3 did not outperform W2 in terms of lipid elimination (*p* > 0.05). The increase in pH value of surimi (Table 1) may have resulted in a decrease in lipid content. According to Chaijan et al. [15], neutral and polar lipids may be eliminated from the mince when the pH was changed away from the pI of myofibrillar proteins, resulting in the release of intra- and inter-muscular lipids.

#### 3.2.3. Myoglobin Content, Derivatives, and Absorption Spectra in the Soret Region

Myoglobin is a heme protein that is responsible for the quality of fish muscle, particularly for the development of color and rancidity. Surimi quality factors (e.g., whiteness, lipid oxidation, microbial growth, and shelf life) are all affected by the occurrence of residual heme proteins in surimi, especially dark-fleshed fish [21,22]. Table 1 shows the myoglobin content and myoglobin derivatives of unwashed mackerel mince and surimi produced using various washing cycles. Myoglobin content in unwashed mince was 6.77 mg/g sample, whereas it was 2.56, 1.63, and 1.61 mg/g sample in surimi after washing with W1, W2, and W3. The W2 and W3 samples had the same residual myoglobin levels (*p* > 0.05). Sarcoplasmic proteins, such as myoglobin, can be dissolved in water, and be washed out of the muscle during water washing [9]. By washing minced fish in cold water, most sarcoplasmic proteins, including myoglobin, hemoglobin, and other undesired compounds, may be eliminated, and the whiteness of surimi can be improved [23]. Parts of myoglobin, on the other hand, have the ability to interact with myofibrillar proteins [7]. The interaction of myoglobin with muscle constituents, such as myofibrillar proteins and/or membranes, can result in impaired extractability, according to Chaijan and Undeland [22]. As a result, increasing the number of washing cycles from two to three did not result in a greater capacity for myoglobin removal.

Deoxymyoglobin (purple red), oxymyoglobin (bright red), and metmyoglobin (tan or brown) are the three major forms of myoglobin, which differ in their exposure to oxygen and the chemical state of iron. W3 sample had the highest amount of deoxymyoglobin, followed by W2 and W1, while unwashed mince had the lowest (*p* < 0.05). In all of the samples, there was no significant variation in the proportion of oxymyoglobin (*p* > 0.05). All of the samples contained a high percentage of oxidized metmyoglobin (~73–76%), which was quite likely to happen during sample handling, processing, and storage. However, increasing the number of washing cycle up to two cycles can lower metmyoglobin levels. Increasing the number of washing cycles can prevent the deoxymyoglobin form of surimi from converting to metmyoglobin, resulting in a delay in lipid oxidation and a reduction in TBARS value (see Section 3.4.4). Deoxymyoglobin and oxymyoglobin can both be oxidized to metmyoglobin. The conversion of reduced myoglobin to oxidized forms resulted in a greyish-brown color in fish muscle, which was related to lipid oxidation [24]. Both a superoxide anion and a hydrogen peroxide are formed during the oxidation of oxymyoglobin, which then react with iron to produce hydroxyl radicals. These hydroxyl radicals have the capacity to permeate the muscle’s hydrophobic lipid area, facilitating lipid oxidation [22,25]. 

The absorption spectra in the Soret region (380–450 nm) of myoglobin extracted from mackerel unwashed mince and surimi are depicted in Figure 1. The content and oxidation state of the myoglobin can be observed by absorption bands in the ultraviolet and visible wavelengths. Anderson and Robertson [26] stated that the heme is responsible for the intense peak at about 410 nm. The peak in the Soret band of all samples was identified at 407 nm, according to the results. Metmyoglobin predominance was revealed by the blue shift from 410 nm to 407 nm. This result confirmed the metmyoglobin content shown in Table 1. During iced storage, Wongwichian et al. [25] observed a shift from 410 nm to 408 nm in oxeye scad (Selar boops) muscle, which was associated with a substantial rise in metmyoglobin concentration. The sample that had not been washed exhibited the highest Soret peak. Surimi made by increasing the number of washing cycles, especially W2 and W3, showed a decrease in Soret peak, demonstrating the removal effectiveness of myoglobin from minced fish.

#### 3.2.4. Ca^2+^-ATPase Activity, Reactive SH Content, and Surface Hydrophobicity

Ca^2+^-ATPase activities and reactive SH contents of NAM extracted from unwashed mince and surimi samples are shown in Table 1. Both biochemical indicators are related to myosin integrity as a result of the washing cycle [27]. The greatest of Ca^2+^-ATPase activity was found in unwashed mince (*p* < 0.05). A marginal decrease in the Ca^2+^-ATPase activities was observed in all surimi (*p* < 0.05), regardless of the number of washing cycles. Das et al. [20] reported that a gradual decrease in the Ca^2+^-ATPase activities was found in sardine, mackerel, pink perch, and croaker surimi, when compared to unwashed mince. The loss of myosin integrity caused by denaturation and/or aggregation was suggested by a decrease in Ca^2+^-ATPase activity of NAM [27]. Loss of Ca^2+^-ATPase activity can be attributed to the structural changes of the myofibrillar proteins during washing process [20].

The presence of SH groups in surimi can promote the formation of disulfide bonds, which is necessary for gel strengthening upon gelation [28]. From results, reactive SH content of surimi tended to increase after washing for one cycle (W1), and the significant increase was found in W2 or W3 treatment (*p* < 0.05). Thus, the washing can enhance the unfolding of protein structure leading to expose of buried SH. According to Buttkus [29], the head and tail sections of the myosin molecule included several reactive SH groups. The reactive SH groups in myosin molecules may become increasingly exposed when their conformation changes [30]. A non-significant difference in SH content with a longer washing cycle could be related to the limited number of SH or the creation of disulfide linkages during washing, resulting in a constant SH content. SH residues are required for disulfide bonding to enhance the gel structure [27,28].

The surface hydrophobicity values of unwashed mince and surimi samples are shown in Table 1. Generally, the hydrophobic amino acid residues are situated internally in the protein molecules [11], and the augmentation in surface hydrophobicity is related to the exposed hydrophobic groups of protein molecules [31]. From the results, the lowest surface hydrophobicity was observed in unwashed mince (*p* < 0.05). For washed treatments, the increasing of washing cycles markedly increased surface hydrophobicity of surimi (*p* < 0.05). Protein unfolding exposes hydrophobic amino acids, changing the hydrophobicity of the protein’s surface [32]. However, a good gel-forming ability of protein needed a proper change on surface hydrophobicity [9]. During thermal gelation, exposed hydrophobic residues can form interactions and strengthen the gel network [7,15].

#### 3.2.5. TCA-Soluble Peptide

TCA-soluble peptide was used to assess protein degradation in unwashed mince and surimi (Table 1). The presence of TCA-soluble peptide indicated proteolytic degradation, and a larger content indicated more muscle protein hydrolysis [30]. Unwashed mince had the greatest TCA-soluble peptide (*p* < 0.05), indicating the occurrence of the most endogenous proteases capable of producing proteolytic breakdown products. Sarcoplasmic proteins in mackerel, including endogenous proteinases and certain soluble peptides, accumulated in the mince due to autolysis during handling and storage. Peptides can be eliminated, and proteases can be partially inactivated by repeatedly washing the mince. Among the surimi, the lowest TCA-soluble peptide was obtained in surimi prepared by W3. Increasing the number of washing cycles for surimi manufacturing was effective in removing some proteinases, resulting in a decrease in the quantities and activity of these enzymes, which was related with less protein breakdown. The soluble peptides can also be eliminated by repeating the washing process. Proteolytic enzymes bind to myofibrillar proteins and interfere with gelation, which is why dark meat fish species have poor gelation capabilities [8]. Activation of myofibril-bound proteases can occur during the initial stages of heat gelation, resulting in the gel-weakening phenomena [8].

### 3.3. Oscillatory Dynamic Rheology

The physicochemical parameters that link to gelation, which is the basis for texture creation, can be determined using rheology [33]. Figure 2 depicts the dynamic viscoelastic behavior of unwashed mackerel mince and surimi generated by varying the number of washing cycles during the temperature transition from sol to gel. Gel formation was measured using the storage modulus (G’) (Figure 2a). An increase in G’ indicated that the sample’s rigidity had increased as a result of the development of an elastic structure [34]. In all samples, the G’ was gradually increased from around 35 °C to approximately 45 °C, as shown in Figure 2a, with the exception of W3, which was elevated continuously. The creation of the initial protein network structure was referred to as the “gel setting” step. Myosin would unfold at this point to allow for ordered polymerization, and the initial elasticity of proteins would be lost [35]. According to Buamard et al. [36], the development of protein networks via weak connections between protein molecules, such as hydrogen bonds, falls within this spectrum. The setting phenomenon may be reduced in W3, which has been washed excessively, resulting in a larger elimination of endogenous transglutaminase (TGase). As TGase is a water-soluble protein, it can be washed away. Following that, G’ dropped rapidly, reaching its lowest point about 48 °C, indicating gel weakening. This was mostly owing to the activity of residual proteases at this temperature, which can hasten myosin degradation, and interfere with gel formation [37]. The majority of heat activated fish proteinases are triggered between 50 and 60 °C, which is close to their optimal temperature [37]. Furthermore, increased surimi paste mobility due to actomyosin dissociation, unfolding, and denaturation can be another explanation for decreased G’ value [31]. Sano et al. [38] also claimed that the drop in G’ was caused by the coil transformation of the α-helix, in which the fluidity of the protein increased and the viscoelasticity decreased; this also explained why G” decreased (Figure 2b). G’ of surimi from Pacific whiting achieved a minimum value of 55 °C, according to Rawdkuen et al. [39]. Nonetheless, the W3 sample did not show a decrease in G’ in this range. This corresponded to the lowest TCA-soluble peptide content (Table 1), owing to the effectiveness of adequate washing, which can remove endogenous protease enzymes. To avoid gel weakening caused by leftover proteases, the surimi preparation was set at 40 °C, and cooked at 90 °C. Myosin denatures and aggregates more at 48 °C, eventually producing a stable gel [40]. The G’ was then increased once again, peaking around 65–69 °C, suggesting the establishment of a strong gel network due to higher attractive forces (e.g., disulfide bond, and hydrophobic interaction). During this “gel strengthening” stage, the primary cross-linking and aggregates of myosin occurred. The unwashed mince provided the lowest G’ throughout the heating process, while the W3 sample had the greatest G’, followed by W2 and W1 samples, respectively, notably at temperatures above 50 °C. This was in line with the same breaking force pattern (see Section 3.4.1). The construction of a thermo-irreversible gel network was most likely owing to the covalent cross-links among dissociated protein molecules. While SH groups were oxidized, a disulfide bridge might have been produced, and hydrophobic domains may have joined via hydrophobic-hydrophobic interaction [41]. G’ was then reduced one more. This could be because hydrogen bonding breaks off at high temperatures [31]. The viscosity modulus (G”) curves of unwashed mince and surimi were often similar to G’ (Figure 2a). The G” followed the same trend as the G’, but the degree of ordered interaction was significantly stronger, resulting in a lower G” value, showing that the elastic composition was the most important factor in surimi gel production. The G” rose until it reached 38 °C, then dropped quickly to its lowest point at 48 °C. Following that, G” climbed steadily up to 65 °C before dropping due to cross-linking [42].

Because G’ values were higher than G” values, the tan δ of all samples was less than 1.0. As a result, all of the samples behaved like an elastic fluid with improved gelation properties [40]. The tan δ represents the distribution of “viscosity” relative to “elasticity”, which is shown by a pure solid at 0° and a pure liquid at 90° [43]. All surimi samples had a first peak at around 36 °C, as shown in Figure 2c. The unwashed sample, on the other hand, performed poorly in this temperature. The unfolding of the myosin structure at temperatures above 30 °C increased the viscosity of the sol [44]. Protein networks were generated as the temperature neared 42 °C (gel setting stage), as a result of the disulfide-linked heavy meromyosin, causing a drop in tan δ [45]. Following that, the tan δ rose rapidly up to 50 °C, which could be due to greater protein unfolding and interaction between unfolded proteins and water molecules at low temperatures (before 50 °C). The heavy meromyosin connected by a disulfide bond was broken by the unfolding of subfragment-2, therefore the tan δ increased as the temperature climbed in the 44–50 °C range. Because the structural modification of myosin had already been performed and the protein structure rearranged, the disulfide linkages and hydrophobic interactions between the molecules were strengthened when the temperature was above 50 °C, resulting in increased rigidity and intensity, but weakened mobility. When the temperature reached above 50 °C, this might be used to explain why the tan δ faded as the temperature rose [46]. The sol-gel transition (tan δ) of washed mince from threadfin bream revealed three transition points, which occurred at 36.7, 43.3, and 70.0 °C, according to Karthikeyan et al. [47]. The transitions at 36.7 and 43.3 °C referred to the myosin tail and the transition at the higher temperature referred to the myosin head [47].

Furthermore, it had depicted shifting of curves (G’ and G”) and peaks (tan δ) at around 50 °C in this study. When compared to the unwashed sample, this was shifted to the lowest temperature during the transition phase of the W3 sample, followed by W2 and W1. It could be a difference in concentration and function of myofibrillar proteins or interfered gelling components, as well as proteins’ ability to entrap water in gel networks, which could be in accordance with the expressible drip of samples (see Section 3.4.2), resulting in interval changes in gel formation. According to Lee and Yoon [48], increasing the moisture content increased the denaturation temperature of fish proteins, and caused the gel formation to be delayed. Despite the fact that the final G’ and G” of surimi differed, the tan δ was nearly same, indicating that all surimi will undergo the same degree of sol-gel transition after thermal gelation.

### 3.4. Gelling Characteristics

#### 3.4.1. Breaking Force, Deformation, and Gel Strength

Effect of washing cycle with cold water on the breaking force, deformation, and gel strength of the surimi gels is illustrated in Table 2. One of the most essential factors in determining the quality and pricing of surimi is the gel strength. Two criteria were used to determine it: breaking force, and deformation. The former referred to surimi gel’s hardness, while the latter referred to its springiness [7]. Unwashed mince gel showed the lowest breaking force and deformation in the current study, while W2 and W3 samples had the highest breaking force (*p* < 0.05). Nonetheless, there was no significant difference in deformation between surimi gels (11 mm) (*p* > 0.05). Furthermore, all surimi had a higher gel strength (breaking force × deformation) than unwashed mince.

Unwashed mince had the highest lipid content (Table 1), and may have more sarcoplasmic protein, which could interfere with the gel network formation during the heat setting process. Furthermore, surimi gel strength increased in tandem with increases in pH, reactive SH content, and surface hydrophobicity, as well as a decrease in TCA-soluble peptide content (Table 1). According to Nowsad et al. [49], increasing the number of washings raises the pH of the hen surimi, and enhances its folding score. Two-step heating is commonly used to produce gel, according to Hossain et al. [50]. Solubilized myofibrillar protein with salt was held at roughly 40 °C to generate an elastic gel before being cooked at >80 °C to form a stronger cooked gel. Interestingly, surimi gel strength from W2 and W3 (~440 g·cm) was higher than the Thai Industrials Standard value for frozen mince fish (not less than 400 g·cm) [51], implying that surimi from Indian mackerel prepared with fewer washing cycles met the standard quality.

#### 3.4.2. Expressible Drip

The expressible drip of gels made from unwashed mince was 8.14%, but after one washing cycle (W1), it dropped to 5.37% (Table 2). Surimi prepared with two (W2) and three (W3) washing cycles (*p* < 0.05) had the lowest expressible drip. This corresponded to surimi gels having a higher gel strength (Table 2). According to Hassan et al. [17], the expressible drip of *Pangasius hypophthalmus* surimi decreased with the increasing in the number of washing cycles. When compared to unwashed mince, the expressible drip of surimi gel from threadfin bream decreased from 29.47% to 19.73%, resulting in an increase in gel strength from 304.39 to 604.47 g·cm, according to Karthikeyan et al. [47].

A good surimi gel is required to create three-dimensional networks of myofibrillar proteins that are stabilized by numerous covalent and non-covalent interactions. The term “expressible drip” is commonly used to characterize surimi gel’s ability to hold water [7]. It is an indication of changes in the molecular structure and electrical charge of myofibrillar proteins, and pH is a critical factor [47]. The ability of surimi gels to hold water may be improved by increasing surimi pH after washing, which is shifted away from the isoelectric point (pI) of myofibrillar proteins (pH ~ 5.5). The agglutination of proteins occurs when the pH is close to pI, resulting in the release of water. After the pH declined to near its pI, the gel-forming ability was reduced [8,9]. The pH of unwashed mince (Table 1) was 6.31, which was near the pI of myofibrillar proteins (pH 5.5). Therefore, the gel’s breaking force and water holding capacity were the lowest. Furthermore, the microstructural characteristics of gel may be linked to its water holding capacity (see Section 3.4.7).

#### 3.4.3. Whiteness

Whiteness, coupled with gel strength, is one of the most essential indications of surimi quality [7]. Unwashed mince had the lowest gel whiteness of 65.11 (Table 2), which could be attributable to the high myoglobin content (Table 1) and the formation of metmyoglobin after heat gelation. The whiteness of the gel could be improved by increasing the washing cycles, according to Hassan et al. [17]. The washing process has the potential to eliminate pigments such as myoglobin, hemoglobin, and melanin, resulting in surimi that is lighter in color. W2 and W3 had the highest whiteness among the surimi gels (*p* < 0.05). It was supported by the myoglobin content (Table 1) and the myoglobin Soret peak (Figure 1). The whiter color of surimi gel could be due to the washing procedure removing more heme pigments and lipid-soluble pigments, such as carotenoids [52]. Lipid oxidation and the Maillard reaction were among factors that could impact the whiteness of unwashed mince gel. Since the lipid in unwashed mince is not removed, it can oxidize during the thermal gelation process. The concentration of aldehydes, which are key secondary lipid oxidation products, can be represented by the TBARS content (Table 2). The Maillard process can be aided by lipid oxidation products, particularly aldehydes, according to Chaijan et al. [53].

#### 3.4.4. Lipid Oxidation and Fishy Odor

Lipid oxidation in muscle foods has a significant negative impact on overall quality. It causes a loss of qualitative attributes such as color, flavor, texture, and nutritional value [22]. The TBARS method is a widely used method for measuring lipid oxidation [54]. Table 2 shows the TBARS contents of gels prepared from unwashed mince and surimi with various washing cycles.

Unwashed gels with the greatest lipid and endogenous prooxidants (e.g., myoglobin) (Table 1) had the highest TBARS content, indicating that lipid oxidation can be enhanced to a greater extent when heated. Surimi gels had a lower content of TBARS than unwashed mince gels (Table 2). Some lipid and endogenous prooxidants that can be oxidized during heating were eliminated by cold water washing. According to Singh et al. [55], fish tissue, particularly dark-fleshed fish, contains iron-binding protein, which can be dissociated during thermal gelation. Free iron may operate as a prooxidant, speeding up the oxidation of lipids. Lower TBARS levels may be obtained as a result of removing lipids and oxidative constituents from mince [56]. In general, lipid and myoglobin oxidation in fish muscle occurred at the same time, and one process seemed to promote the other [22,25]. W2 and W3 had the same TBARS level, which was found to be the lowest content (*p* < 0.05) in the study. This was most likely owing to the identical lipid and myoglobin contents found in W2 and W3 surimi (Table 1).

Table 2 shows the fishy odor scores of unwashed mince and surimi gels. Undoubtedly, the gel from the unwashed mince had the strongest fishy odor (*p* < 0.05). The findings were consistent with Phetsang et al. [57], who discovered that unwashed hybrid catfish mince had a stronger fishy odor than conventional surimi. After washing, the fishy odor seemed to reduce as the number of washing cycles increased. However, sensory trained panelists did not perceive a significant difference in the fishy odor of surimi after two (W2) and three (W3) washing cycles. As a result, two-cycle washing can be utilized to make mackerel surimi with a good odor score.

#### 3.4.5. FTIR Spectra

The FTIR spectra of gels of unwashed mince and surimi washed with different washing cycles are shown in Figure 3. The FTIR technique is widely used to figure out a protein’s secondary structure [5]. There was a decrease in α-helix structures and an increase in β-sheet structures after heating for creating unwashed mince/surimi gel, which was attributable to salt solubilization of protein [58]. During the manufacture of surimi sol, approximately 2.5 % (*w*/*w*) salt was added prior to heating to be gelled [7]. Amide III, II, I, and amide B/A vibrational modes in protein molecule are indicated by absorptions in the infrared ranges of 1200, 1500, 1700, and 3100–3300 cm^−1^, respectively [59]. The identical spectra were observed in all samples in Figure 3, showing that the gel structure contained the same protein functional groups. In the FTIR spectra, the primary bands were identified as amide I, II, III, A, and B. At 1640 cm^−1^, a significant amide I peak in the spectra of all gels generated from the C=O stretching vibration, and extensively employed as a probe for protein conformation, was found, indicating a supremacy of the β-sheet and thus a more stable structure [60]. N–H bending and C–N stretching vibrations were attributed to the amide II band, which occurred at 1530 cm^−1^. At 1236 cm^−1^ (N–H deformation and C–N stretching vibration), amide III can be found. The N−H stretching vibration of amide A was measured at 3273 cm^−1^, while the CH stretching vibrations of –CH_2_ group of amide B were measured at 2925 cm^−1^. Increasing the number of washing cycles with cold tap water did not appear to impact the chemical makeup of myofibrillar proteins.

#### 3.4.6. SDS-PAGE Protein Patterns

Figure 4 shows the SDS-PAGE protein patterns of unwashed mince, surimi, and their gels. The predominant bands were determined to be myosin heavy chain (MHC) at 205 kDa and actin at 45 kDa. Myosin is the major protein responsible for the development of the gel network among them [61,62,63]. In the absence of βME in all samples, some polymerized proteins populated the stacking gel in the non-gelled samples (Figure 4a), suggesting the sensitivity of mackerel muscle proteins to aggregation during handling and surimi production. Most polymerized proteins can be dissociated in the presence of βME. However, a part of the protein aggregates remained in the stacking gel, suggesting the presence of other bonds, such as hydrophobic interactions. When compared to surimi, the unwashed mince revealed some protein bands below 25 kDa, indicating the existence of sarcoplasmic proteins in this sample. Surimi, on the other hand, showed no noticeable changes in MHC band intensity. Myosin is a salt-soluble protein family. As a result, it can be retained after being washed with water. When thorough washing (e.g., 4-cycle water washing) was used, however, some myosin was lost [17].

MHC bands intensity decreased after gelation (Figure 4b) in the presence and absence of βME, relative to that seen in the original unwashed mince and surimi, presumably to polymerization during heating. The reduction in MHC bands was thought to be caused by protein cross-linking during setting, which resulted in reduced band intensity [61]. The result was accompanied by an increase in gel strength (Table 2). In all of the samples, there were no discernible changes in actin bands. Actin (45 kDa) was shown to be the main protein in the gel in the current investigation, while Balange and Benjakul [63] made a similar observation, reporting that actin could not be polymerized during gelation. In general, actin may not be a TGase substrate, and the tropomyosin and troponin fractions did not contribute to the gel formation, hence their levels did not alter after the gel was formed [64]. In the absence of βME, the polymerized proteins appeared and hung on the stacking gel for all gels (Figure 4b), but their intensities were reduced in the presence of βME. According to the findings, all of the gels were stabilized by the disulfide bond as well as other bonds such as hydrophobic interaction.

#### 3.4.7. Microstructure

The microstructures of the unwashed mince gel and surimi gels prepared by different of washing cycle are shown in Figure 5. The gel samples all displayed a network structure, showing that the gels were elastic in nature. Unwashed mince gel featured porous networks, with numerous microscopic gaps evenly dispersed throughout the gel structure. Furthermore, the crosslinking of multiple fibrous proteins was detected, resulting in a spongy structure with small fibrous networks. This was owing to the existence of various components in unwashed mince, such as sarcoplasmic proteins and lipids, which have varying gel-forming abilities. Sarcoplasmic proteins can bind to myofibrillar proteins and disrupt the creation of a strong gel network [7,65]. Retained lipids have been shown to have a deleterious impact on the gel strength of tropical fish surimi. Because lipids lack the ability to form a gel matrix, they most likely disrupted the connection between protein molecules [7,57]. Surimi gels (W1–W3) that were subjected to varied washing cycles, on the other hand, had very comparable microstructures, particularly the W2 and W3 samples. Due to the influence of encapsulated capillary water, some concentrated dense region and distributed pores of varying size and depth were created by myosin gel networks, resulting in better strength, and some larger size, deep and localized holes. This indicates that surimi has a larger water retention capacity (Table 2). The wider and deeper pores occurred as a result of trapped water within the cooked gel networks, according to Bertram et al. [66].

## 4. Conclusions

The bio-physico-chemical characteristics of Indian mackerel surimi can be improved by increasing the number of washing cycles. W2 and W3 were found to be the best washing cycles, as they provided the most gel strength, water holding capacity, whiteness, and less lipid oxidation, as well as fishy odor. W2, on the other hand, created a higher surimi yield than W3. As a result, two washing cycles may not only be excellent for producing high-quality surimi from Indian mackerel, but it may also reduce water consumption dramatically. This research will aid in determining the feasibility of Indian mackerel as a raw material for surimi production, which will be beneficial to the surimi industry’s sustainability.

## Figures and Tables

**Figure 1 foods-10-02717-f001:**
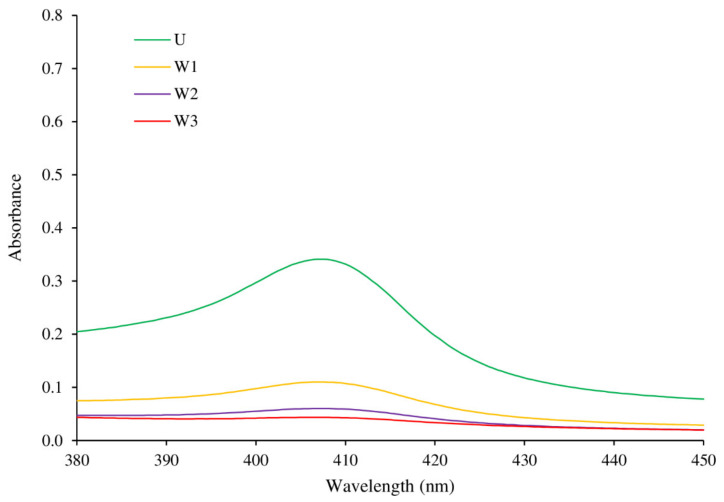
Changes in absorption spectra in the Soret region (380–450 nm) of myoglobin extracted from unwashed mince (U) and surimi from Indian mackerel prepared by one washing cycle (W1), two washing cycles (W2), and three washing cycles (W3).

**Figure 2 foods-10-02717-f002:**
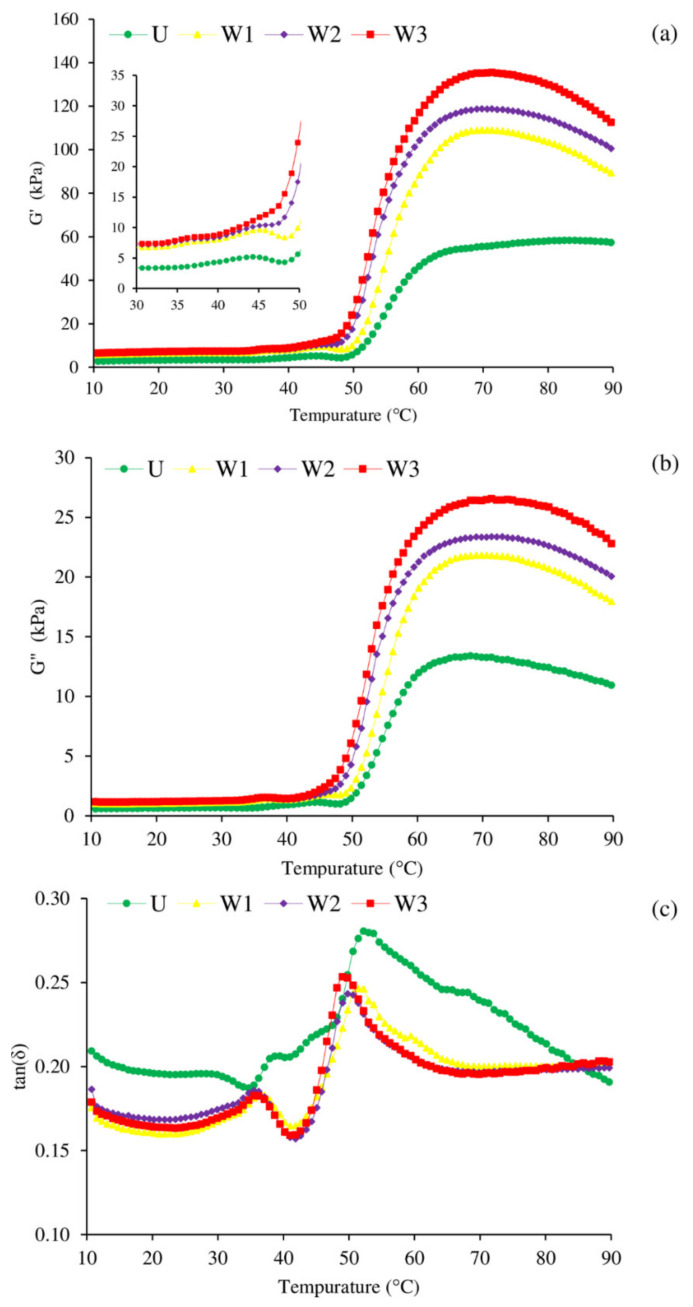
Changes in dynamic viscoelastic behavior of unwashed mince (U) and surimi pastes from Indian mackerel prepared by one washing cycle (W1), two washing cycles (W2), and three washing cycles (W3). The rheograms show elastic modulus, G’ (**a**), viscous modulus, G” (**b**), and tan δ (**c**). Samples were heated from 10 °C to 90 °C.

**Figure 3 foods-10-02717-f003:**
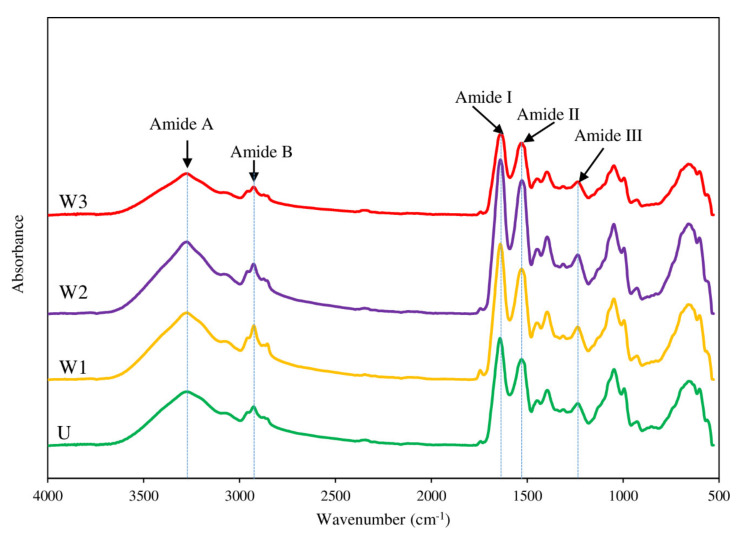
FTIR spectra of gels from unwashed mince (U) and surimi from Indian mackerel prepared by one washing cycle (W1), two washing cycles (W2), and three washing cycles (W3).

**Figure 4 foods-10-02717-f004:**
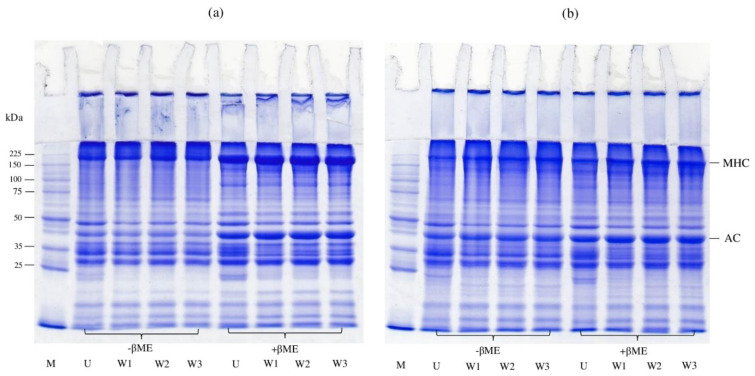
SDS-PAGE patterns before (**a**) and after gelation (**b**) of proteins from unwashed mince (U) and surimi from Indian mackerel prepared by one washing cycle (W1), two washing cycles (W2), and three washing cycles (W3). MHC, myosin heavy chain; AC, actin; M, protein markers; −βME, without β-mercaptoethanol; +βME, with β-mercaptoethanol. Samples (15 μg protein) were loaded onto the polyacrylamide gels comprising a 10% running gel and a 4% stacking gel, and subjected to electrophoresis at a constant current of 15 mA/gel.

**Figure 5 foods-10-02717-f005:**
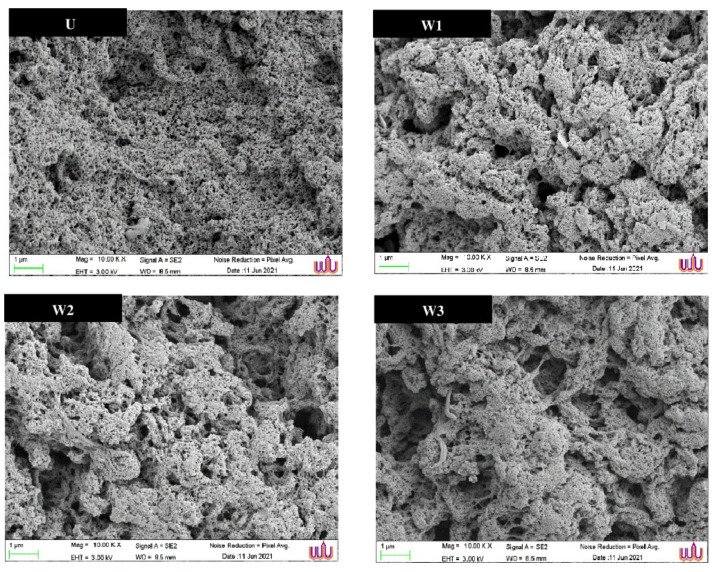
Scanning electron microscopic (SEM) images of gels from unwashed mince (U) and surimi from Indian mackerel prepared by one washing cycle (W1), two washing cycles (W2), and three washing cycles (W3). (Magnification: 10,000×, EHT: 3 kV).

**Table 1 foods-10-02717-t001:** Yield and bio-physico-chemical properties of Indian mackerel unwashed mince and surimi prepared by different washing cycles.

Sample	Unwashed	W1	W2	W3
Yield (%)	-	89.57 ± 4.32a	83.43 ± 3.77b	79.74 ± 2.46c
pH	6.31 ± 0.01d	6.45 ± 0.01c	6.62 ± 0.01b	6.75 ± 0.01a
Lipid reduction (%)	-	52.76 ± 1.31b	54.36 ± 0.76ab	56.43 ± 1.24a
Myoglobin content (mg/g sample)	6.77 ± 0.72a	2.56 ± 0.15b	1.63 ± 0.01c	1.61 ± 0.06c
Myoglobin derivatives				
Deoxymyoglobin (%)	17.59 ± 0.23d	18.92 ± 0.33c	20.96 ± 0.85b	22.56 ± 0.32a
Oxymyoglobin (%)	6.29 ± 0.09a	5.23 ± 0.65a	5.27 ± 0.65a	4.61 ± 0.76a
Metmyoglobin (%)	75.12 ± 0.13a	75.85 ± 0.92a	73.78 ± 0.78b	72.83 ± 0.45b
Ca^2+^-ATPase activity (µmol/mg protein/min)	2.27 ± 0.03a	2.17 ± 0.04b	2.12 ± 0.03b	2.10 ± 0.02b
Reactive SH content (mol/10^8^ g protein)	6.14 ± 0.05b	6.29 ± 0.02ab	6.48 ± 0.09a	6.49 ± 0.20a
Surface hydrophobicity (BPB bound (µg))	30.43 ± 1.42d	41.42 ± 2.48c	49.43 ± 2.86b	57.61 ± 0.33a
TCA-soluble peptide content (µmol tyrosine/g sample)	0.79 ± 0.03a	0.43 ± 0.05b	0.31 ± 0.03c	0.19 ± 0.03d

Values are given as mean ± standard deviation from triplicate determinations. Different letters in the same row indicate significant differences (*p* < 0.05). W1 = one washing cycle, W2 = two washing cycles, W3 = three washing cycle, and BPB = bromophenol blue. Lipid content of unwashed mince was 0.61%, wet weight. In the unwashed mince, the yield and lipid reduction were not calculated.

**Table 2 foods-10-02717-t002:** Gelling properties of Indian mackerel unwashed mince and surimi prepared by different washing cycles.

Sample	Unwashed	W1	W2	W3
Appearance	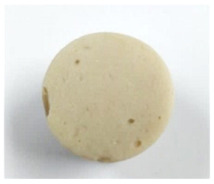	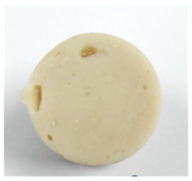	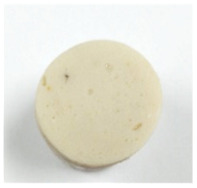	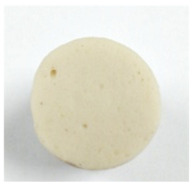
Breaking force (g)	218.12 ± 0.44c	353.60 ± 1.93b	393.98 ± 1.53a	393.09 ± 1.81a
Deformation (mm)	10.06 ± 0.36b	10.83 ± 0.34a	11.20 ± 0.58a	11.26 ± 0.31a
Gel strength (g.cm)	219.32 ± 3.67c	382.71 ± 17.04b	436.32 ± 25.23a	442.17 ± 23.97a
Expressible drip (%)	8.14 ± 0.40c	5.37 ± 0.04b	4.63 ± 0.19a	4.51 ± 0.20a
Whiteness	65.11 ± 0.18c	66.44 ± 0.14b	68.43 ± 0.31a	68.98 ± 0.42a
TBARS (mg MDA equivalent/kg)	0.73 ± 0.04a	0.67 ± 001b	0.59 ± 0.01c	0.57 ± 0.04c
Fishy odor score *	9.03 ± 0.45a	4.43 ± 1.12b	2.27 ± 1.17c	1.37 ± 0.98c

Values are given as mean ± standard deviation from triplicate determinations except for fishy odor scores (*n* = 10). Different letters in the same row indicate significant differences (*p* < 0.05). W1 = one-washing cycle, W2 = two-washing cycle, W3 = three-washing cycle, and MDA = malondialdehyde. * A score of 0 represented “none” while 10 represented “strong”.

## Data Availability

Not applicable.

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
