# Peer review of "Reduced Washing Cycle for Sustainable Mackerel (Rastrelliger kanagurta) Surimi Production: Evaluation of Bio-Physico-Chemical, Rheological, and Gel-Forming Properties"

_foods, 2021, doi:10.3390/foods10112717_

Round 1
Reviewer 1 Report
Comments
Even though the manuscript "Reduced Washing Cycle for Sustainable Mackerel (Rastrelliger 2 kanagurta) Surimi Production: Evaluation of Bio-physico-3 chemical, Rheological, and Gel-forming Properties" could be of interest to the Foods readers. The abstract, introduction and discussion should be largely improved.
Line 14-27, The summary is not supported by data conclusions.
Line 15, Grammatical error. Suggest: “reduced” change it to “reducing”
Line 37-38, Is this sentence grammatical?
Line84-86, How to ensure the final moisture content to 80%? What method was used to determine this?
Line 99-101、113、119, Since this experimental method is not mentioned in the previous section, please briefly describe the experimental steps in the cited literature.
Line 160-163, The conclusions here do not cite literature to support their results.
Line 210, The blue shift of 3nm does not seem to have a significant advantage, is there relevant literature support?
Line 227-230, Figure1 The legend's label and the line in the icon do not match, the legend is a dashed line, the line is a solid line.
2.1 Chemicals, How about the purity of the chemicals?
Line 420, This sentence is not very clear. Where is the source of aldehydes? Is it the product of lipid oxidation? Please explain it.
The lines in Figure 1 are too thick.
Please cite some more literatures to support your point during the discussion.
Author Response
Reviewer 1
Even though the manuscript "Reduced Washing Cycle for Sustainable Mackerel (Rastrelliger 2 kanagurta) Surimi Production: Evaluation of Bio-physico-3 chemical, Rheological, and Gel-forming Properties" could be of interest to the Foods readers. The abstract, introduction and discussion should be largely improved.
Line 14-27, The summary is not supported by data conclusions.
Ans: The abstract was revised.
Line 15, Grammatical error. Suggest: “reduced” change it to “reducing”
Ans: This word was changed following the suggestion.
Line 37-38, Is this sentence grammatical?
Ans: We would like the keep this sentence because Quillbot, a paraphrasing tool, was used to double-check the English.
Line84-86, How to ensure the final moisture content to 80%? What method was used to determine this?
Ans: We stated in the Section 2.3 Surimi and Surimi Gel Preparation that “…… After that, the mince was dewatered with a hydraulic press, and the moisture content of the unwashed mince and surimi was assessed using the AOAC method [11]. Unwashed mince had an initial moisture content of about 75%, while surimi had a moisture content of 78-80%. Using chilled water, the final moisture content of the unwashed mince and surimi was adjusted to an average of 80%, and the moisture level was determined again to ensure an 80% final moisture content…….”
Line 99-101、113、119, Since this experimental method is not mentioned in the previous section, please briefly describe the experimental steps in the cited literature.
Ans: As recommended, the methodology was extended and detailed.
Line 160-163, The conclusions here do not cite literature to support their results.
Ans: The literatures were cited.
Line 210, The blue shift of 3nm does not seem to have a significant advantage, is there relevant literature support?
Ans: The formation of metmyoglobin can be influenced by a 3-nm change. Since just a 2-nm change has been shown to affect the content of metmyoglobin. This statement was originally supported by a reference. “Wongwichian et al. [26] observed a shift from 410 nm to 408 nm in oxeye scad (Selar boops) muscle, which associated with a substantial rise in metmyoglobin concentration.”
Line 227-230, Figure1 The legend's label and the line in the icon do not match, the legend is a dashed line, the line is a solid line.
Ans: Thank you very much. Figure 1 was fixed as recommended.
2.1 Chemicals, How about the purity of the chemicals?
Ans: All chemicals were of analytical grade. It was noted in manuscript
Line 420, This sentence is not very clear. Where is the source of aldehydes? Is it the product of lipid oxidation? Please explain it.
Ans: The explanation has now been added. “Lipid oxidation and the Maillard reaction were among factors that could impact the whiteness of unwashed mince gel. Because the lipid in unwashed mince is not removed, it can oxidize during the thermal gelation process. The concentration of aldehydes, which are key secondary lipid oxidation products, can be represented by the TBARS content (Table 2). The Maillard process can be aided by lipid oxidation products, particularly aldehydes, according to Chaijan et al. [53].”
The lines in Figure 1 are too thick.
Ans: Figure 1 was fixed as recommended.
Please cite some more literatures to support your point during the discussion.
Ans: The references were cited extensively throughout the article, comprising 64 documents.

Reviewer 2 Report
This manusvript explained the use of mackerel surimi with reduced washing cycle. I felt that the concept(introdution) and conclusion is not completely fitting. I put small comments to conclusion.
Concerning the originality of this work, there are many reports related to R.kanagurta surimi until now. Especially this work is quite similar to the works published from Benjakul group(data and small parts of conditions are different). Authors should explain how this current work is important to publish more precisely. The results showed in the manuscript are valuable, but explanation to express importance is missing.
The figures should be revised, symbols, colors in all figures should be synchronized.
Author Response
Reviewer 2
This manusvript explained the use of mackerel surimi with reduced washing cycle. I felt that the concept(introdution) and conclusion is not completely fitting. I put small comments to conclusion.
Concerning the originality of this work, there are many reports related to R.kanagurta surimi until now. Especially this work is quite similar to the works published from Benjakul group(data and small parts of conditions are different). Authors should explain how this current work is important to publish more precisely. The results showed in the manuscript are valuable, but explanation to express importance is missing.
Ans: Surimi from the dark-fleshed fish R. kanagurta, one of the most abundant species in the Gulf of Thailand, was the subject of this manuscript. Of course, the Benjakul group (which includes Manat Chaijan) has published papers on surimi from this species, but the research questions were rather different. The peculiarity of our study is that we are the first to emphasize the reduced washing cycle for sustainable surimi production from this species. The surimi's bio-physico-chemical, rheological, and gel-forming capabilities were all thoroughly investigated.
The significance was established in the opening sentence of the Abstract as well as in the Introduction. We started with the challenges of employing dark-fleshed fish as a raw material in the Introduction. The second paragraph explained how to make sustainable surimi by decreasing the washing cycle and washing minced fish with cold water, which is a chemical-free method. However, there is little information on the impact of the number of water washing cycles on the bio-physico-chemical, rheological, and gelling properties of R.kanagurta surimi. With this in mind, the purpose of this study was to see how the quality of R.kanagurta surimi was influenced by a reduced number of washing cycles. This research will aid in determining the feasibility of R.kanagurta as a raw material for surimi production, which will be beneficial to the surimi industry's sustainability.
The figures should be revised, symbols, colors in all figures should be synchronized.
Ans: Figures were rechecked as recommended.

Round 2
Reviewer 1 Report
Althoughauthors havemade some changes in the manuscript, but I did not see any response to the discussion and conclusion parts, I think these parts still need to be improved.
Author Response
Thank you very much. The discussion and conclusion parts were improved as suggested. The references were also updated.